# Climate change is associated with higher phytoplankton biomass and longer blooms in the West Antarctic Peninsula

Afonso Ferreira [1,2] ✉, Carlos R. B. Mendes [2,3,4], Raul R. Costa [2,3,4], Vanda Brotas [1,5,6], Virginia M. Tavano [2], Catarina V. Guerreiro [1,5], Eduardo R. Secchi [4,7] & Ana C. Brito [1,5]

The Antarctic Peninsula (West Antarctica) marine ecosystem has undergone substantial changes due to climate-induced shifts in atmospheric and oceanic temperatures since the 1950s. Using 25 years of satellite data (1998-2022), this study presents evidence that phytoplankton biomass and bloom phenology in the West Antarctic Peninsula are significantly changing as a response to anthropogenic climate change. Enhanced phytoplankton biomass was observed along the West Antarctic Peninsula, particularly in the early austral autumn, resulting in longer blooms. Long-term sea ice decline was identified as the main driver enabling phytoplankton growth in early spring and autumn, in parallel with a recent intensification of the Southern Annular Mode (2010-ongoing), which was observed to influence regional variability. Our findings contribute to the understanding of the complex interplay between environmental changes and phytoplankton responses in this climatically key region of the Southern Ocean and raise important questions regarding the far-reaching consequences that these ecological changes may have on global carbon sequestration and Antarctic food webs in the future.

Marine ecosystems along the coast of the Antarctic Peninsula (West Antarctica; Fig. 1) have been undergoing significant change over the past few decades[1–3]. Among the most important is the rise of atmospheric and oceanic temperatures due to global warming, particularly during the winter[1,4], which has led to the generalised retreat of glaciers, ice shelves and to a decrease in the extent, thickness, and volume of sea ice[5–9]. All marine trophic levels have been impacted by these changes[2,3,10,11]. Concerning primary producers, alterations in phytoplankton biomass, composition, and cell size have been reported, with potential consequences for the Antarctic food web[2,6,12–17]. Nevertheless, phytoplankton communities appear to be affected differently along the Western Antarctic Peninsula (WAP)[2,12–15]. In the southern-mid WAP (south of Anvers Island; 64°33'S), studies have shown increases in phytoplankton biomass and cell size[2,14,15,17], while in the northern Peninsula, the current consensus is that phytoplankton biomass and cell size are decreasing[2,12,13,18]. However, it should be noted that the phytoplankton dynamics in the northern part of the Peninsula have been less studied overall and are currently much less understood than the southern-mid WAP[12].

[1]MARE - Marine and Environmental Sciences Centre/ARNET—Aquatic Research Network, Faculdade de Ciências, Universidade de Lisboa, Campo Grande 016, 1749–016 Lisboa, Portugal. [2]Laboratório de Fitoplâncton e Microorganismos Marinhos, Universidade Federal do Rio Grande-FURG, Av. Itália, Km 8, 96203–900 Rio Grande-RS, Brasil. [3]Laboratório de Estudos dos Oceanos e Clima, Universidade Federal do Rio Grande-FURG, Av. Itália, Km 8, 96203–900 Rio Grande-RS, Brasil. [4]Programa de Pós-graduação em Oceanografia Biológica, Universidade Federal do Rio Grande-FURG, Av. Itália, Km 8, 96203–900 Rio Grande-RS, Brasil. [5]Departamento de Biologia Vegetal, Faculdade de Ciências, Universidade de Lisboa, Campo Grande 016, 1749–016 Lisboa, Portugal. [6]Plymouth Marine Laboratory, Prospect Place, Plymouth PL1 3DH, United Kingdom. [7]Laboratório de Ecologia e Conservação da Megafauna Marinha, Universidade Federal do Rio Grande-FURG, Av. Itália, Km 8, 96203–900 Rio Grande-RS, Brasil. ✉e-mail: ambferreira@ciencias.ulisboa.pt

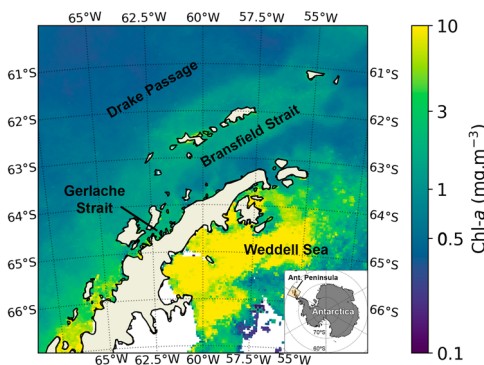

**Fig. 1 | Phytoplankton biomass distribution in the Antarctic Peninsula.** Key areas within the Antarctic Peninsula region and mean spatial distribution of chlorophyll-a (Chl-*a* mg m$^{-3}$) during the full observation period (1998-2022).

While the continuous in-situ observation of marine phytoplankton in the WAP has been undertaken since the second half of the 20th century[19], the latter was mostly restricted to coastal sites along the WAP in the proximity of active research stations and to oceanographic expeditions during the austral summer. This raises difficulties in understanding whether rapid biological changes are already taking place in the region. Ocean colour remote sensing provides a complementary tool to in-situ sampling, which has already proven to crucially contribute to the unravelling of short- and long-term phytoplankton patterns in the ocean at both regional and global scales[20]. Nonetheless, applications of satellite data to Antarctic coastal waters have been hampered by limited data availability due to persistent cloud cover, sea ice, and low solar angles, inter alia[21,22]. Additionally, global satellite algorithms have consistently underestimated chlorophyll-a (Chl-*a*; a proxy for phytoplankton biomass) in the Antarctic Peninsula by up to a factor of 2[22–25].

In this study, we implement the new OC4-SO regional algorithm for Chl-*a*, which was specifically calibrated for the Antarctic Peninsula using the largest high-quality in-situ regional dataset[22] to advance existing knowledge of phytoplankton dynamics and the impact of climate change on primary production in this critical Southern Ocean region. We use 25 years of continuous multi-sensor remote sensing data (ESA OC-CCI[26]) to disentangle patterns of phytoplankton biomass and bloom phenology throughout the Antarctic Peninsula marine ecosystem in relation to regional atmospheric and oceanographic changes related to ongoing climate change. We observe enhanced phytoplankton biomass along the WAP, particularly in the early austral autumn, resulting in longer blooms. Long-term sea ice decline appears to be the main driver enabling growth in early spring and autumn, in parallel with a recent intensification of the Southern Annular Mode (SAM; 2010-ongoing).

## Results and discussion
### Phytoplankton biomass dynamics along the Antarctic Peninsula
Five marine subregions with distinctive seasonal phytoplankton patterns along the Antarctic Peninsula were identified after applying a hierarchical clustering analysis to key phytoplankton biomass and bloom phenology metrics (Fig. 2). From north to south: (i) DRA, which includes waters in the southern Drake Passage, north of the Southern Antarctic Circumpolar Current Front (SACCF; Fig. 2a), (ii) BRS, which extends from Elephant Island to the offshore waters south of Anvers Island, including the Bransfield Strait and the South Shetland Islands (Fig. 2b), (iii) WED$_N$, which is situated between the Bransfield Strait and the outer NW Weddell Sea, off the Northern Antarctic Peninsula (Fig. 2c), (iv) GES, which contains the coastal waters stretching south from the Gerlache Strait (Fig. 2d), and (v) WED$_S$, the NW Weddell coastal waters east of the Antarctic Peninsula, within the Larsen A and B embayments (Fig. 2e).

Phytoplankton blooms off the Antarctic Peninsula generally began in late October or November, with peaks in productivity occurring between December and February and ending in late March or April, coastal locations showing later blooms and higher Chl-*a* peaks (Fig. 2; Supplementary Table 1). DRA, the northernmost and most open ocean subregion, was where phytoplankton started to bloom earlier, right after austral winter. The low biomass levels and early short-lived biomass peak observed in this subregion suggest that micronutrient limitation (potentially iron) may occur shortly after December, corroborating previous in-situ studies that have reported lower biomass in the southern Drake Passage associated with iron limitation[27,28], a characteristic of the more oceanic waters off the WAP during summer[29]. The nature of phytoplankton dynamics among subregions is also related to their physical environment. Regions within the WAP exhibited generally warmer waters, lower sea ice coverage, and stronger winds than the regions in the eastern sector of the Antarctic Peninsula (Supplementary Fig. 1; Supplementary Table 2). Sea ice concentration was the primary factor influencing the timing of bloom initiation around the Peninsula (Supplementary Fig. 2; *p* value < 0.05), as regions with more sea ice during spring were often associated with later blooms (e.g., WED$_N$, GES, and WED$_S$). The importance of environmental conditions in shaping phytoplankton dynamics was also evidenced in BRS, as seen by its high variability in both bloom phenology and biomass. Despite its relatively low sea ice concentration, the Bransfield Strait, at the centre of BRS, exhibits a complex current system that includes cyclonic and anticyclonic eddies, surface and subsurface thermal fronts, as well as intrusions of warmer waters from the west (the Circumpolar Deep Water; CDW) and colder waters from the northeast (from the NW Weddell Sea)[30–33], an overall complexity that translates to high spatial and temporal variability.

### Enhanced biomass and longer blooms in the WAP
Results from a linear trend analysis indicated that the mean phytoplankton biomass from September to April (austral spring to autumn) has significantly increased between 1998-2022 in the WAP, particularly in the Bransfield Strait (*p* value < 0.05; Fig. 3a, Supplementary Fig. 3). At the same time, it is not clear from our results whether biomass also changed in the eastern sector of the Antarctic Peninsula, which may be a consequence of the low number of satellite observations in this sector and lower accuracy caused by the high sea ice coverage[34]. The phytoplankton biomass increase observed in the WAP seems to have occurred mainly in the early spring and early autumn (Table 1). This is evidenced by the increases of +0.003 mg m$^{-3}$ Chl-*a* year$^{-1}$ in the DRA and of +0.009 mg m$^{-3}$ Chl-*a* year$^{-1}$ in the BRS during September (early spring). This was even more the case during early autumn (March and April), when biomass trends were nearly three times higher compared with the early spring: +0.009 mg·m$^{-3}$ Chl-*a* year$^{-1}$ in DRA, +0.026 mg m$^{-3}$ Chl-*a* year$^{-1}$ in BRS, and +0.061 mg m$^{-3}$ Chl-*a* year$^{-1}$ in GES. Therefore, most of the biomass increase in the WAP occurred in early autumn, which corroborates the higher average autumn biomass observed in 2011–2020 compared to 2001–2010 (Fig. 2a, b, d).

The increase in biomass seen in the GES partially corroborates the findings of previous in-situ and satellite studies, although for different temporal periods[2,14–17,35]. For instance, Moreau et al[17]. observed an increase in the annual integrated primary production from 1997 to 2010 using exclusively satellite data[17] but found no clear trend in satellite Chl-*a*. These authors attributed this increase in primary production to the long-term trend towards earlier sea ice retreats and its effects on phytoplankton growth during the spring[17]. Therefore, they did not consider the large phytoplankton growth in autumn that we observed here, which cannot be related to the earlier sea ice retreat. However, Turner et al. using satellite Chl-a from 1997 to 2022, also observed an increase in phytoplankton biomass in the autumn in northern WAP[16]. Previous studies using in-situ data, such as Montes-Hugo et al[2]. (from 1978 to 2006) and Brown et al.[15] (from 1993 to 2017),

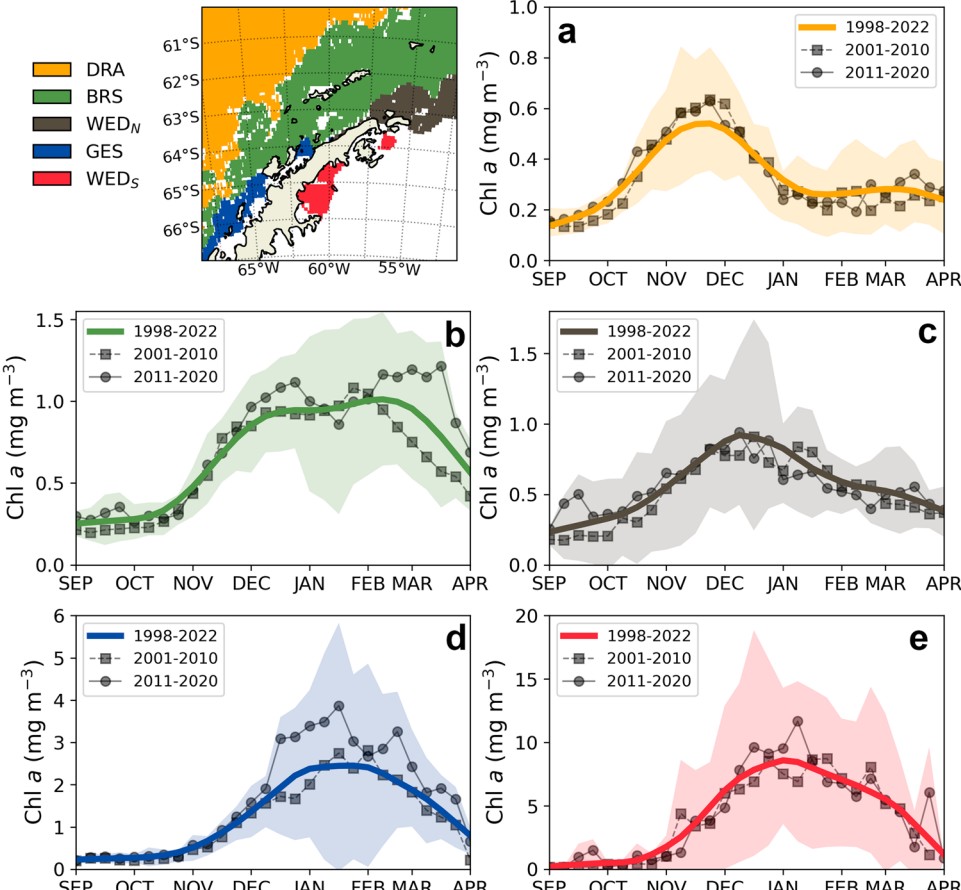

**Fig. 2 | Seasonal cycles of phytoplankton biomass across regions of the Antarctic Peninsula.** Location of regions with coherent chlorophyll-*a* (Chl-*a*; mg m$^{-3}$) seasonal cycles stemming from the hierarchical clustering analysis and smoothed mean seasonal cycle during the full observation period (1998–2022; solid lines), 2001–2010 (lines with squares), and 2011–2020 (lines with circles) for each region. **a** DRA. **b** BRS. **c** WED$_N$. **d** GES. **e** WED$_S$. **a**–**e** The yellow, green, grey, blue, and red background shades represent the mean of the seasonal cycle ± the standard deviation for each region, respectively.

also observed a similar decadal enhancement of phytoplankton biomass in this region, yet, unlike our satellite-detected trends, they only observed it for the summer (the only period for which the current availability of in situ data allows for more accurate long-term trend assessments). These studies suggested that this biomass increase was a consequence of a more stable upper mixed layer during the summer due to the meltwater resulting from climate-induced sea ice decline and glacier retreat[2,15]. While our region-wise satellite observations do not show a clear increasing trend of biomass during summer in the mid/southern WAP (GES), the average summer biomass between 2011 and 2020 does appear to have increased in comparison to the previous decade (Fig. 2d). Moreover, a slight increasing biomass trend was also observed for December (although only at a *p* value = 0.09; Table 1) and pixel-wise trend analyses for the summer do suggest that biomass might be increasing in coastal waters south of the Gerlache Strait, although the significant increase is locally observed (Supplementary Fig. 3). A reanalysis of in-situ Chl-*a* data collected along the WAP (from Palmer LTER and the Brazilian High Latitude Oceanography Group) also suggests that summer biomass has increased in the BRS (+ 0.06 m ·m$^{-3}$ year$^{-1}$) and GES (+ 0.18 mg m$^{-3}$ year$^{-1}$) (Fig. 3b; Supplementary Fig. 4). Therefore, there may have been, in truth, an increase in summer biomass that satellite data did not accurately capture.

Our findings also help to shed some light on the current understanding of the northern WAP (DRA and BRA regions). Our results contradict the hypothesis proposed by Montes-Hugo et al[2]. in 2009, who suggested that regional phytoplankton biomass is decreasing due

to the deepening of the upper mixed layer caused by less sea ice concentration and enhanced wind stress[2], although it is important to note that this study compared the period between 1978–1986 and 1998–2006. Other phytoplankton ecology studies have since echoed this hypothesis, although no further study has confirmed it using long-term datasets. Here, however, we demonstrate that phytoplankton biomass is increasing since 1998, an increase concurrent with wind stress (Supplementary Fig. 5), which suggests that phytoplankton growth has not been hindered by a long-term deepening of the mixed layer resulting from stronger winds[35]. Our observations is supported by recent studies reporting localized long-term increases in summer biomass in Potter Cove (King George Island; 62.2382°S, 58.6673°W) and the region between the South Shetland Islands and the Elephant and Clarence islands, although neither of these works report increases in autumn biomass[36,37]. Contrarily, Turner et al. observed a recent increase in autumn biomass in the northernmost waters of the WAP, yet did not report long-term increase in annual Chl-a[16].

The changes in phytoplankton biomass have led to changes in bloom phenology (Table 2), with blooms ending later in two of the three studied sub-regions of the WAP (+ 0.4 weeks year$^{-1}$ in GES and +0.1 weeks year$^{-1}$ in DRA). As a result, blooms have become longer in both regions, thus resulting in higher total biomass production, particularly at the DRA (*p* value < 0.05). While the timing of the bloom termination changed, there were no apparent shifts in the timings of bloom initiation or bloom peak. Surprisingly, no trends were observed for BRS despite also being a region where biomass increased significantly in the autumn. This was likely due to the high interannual

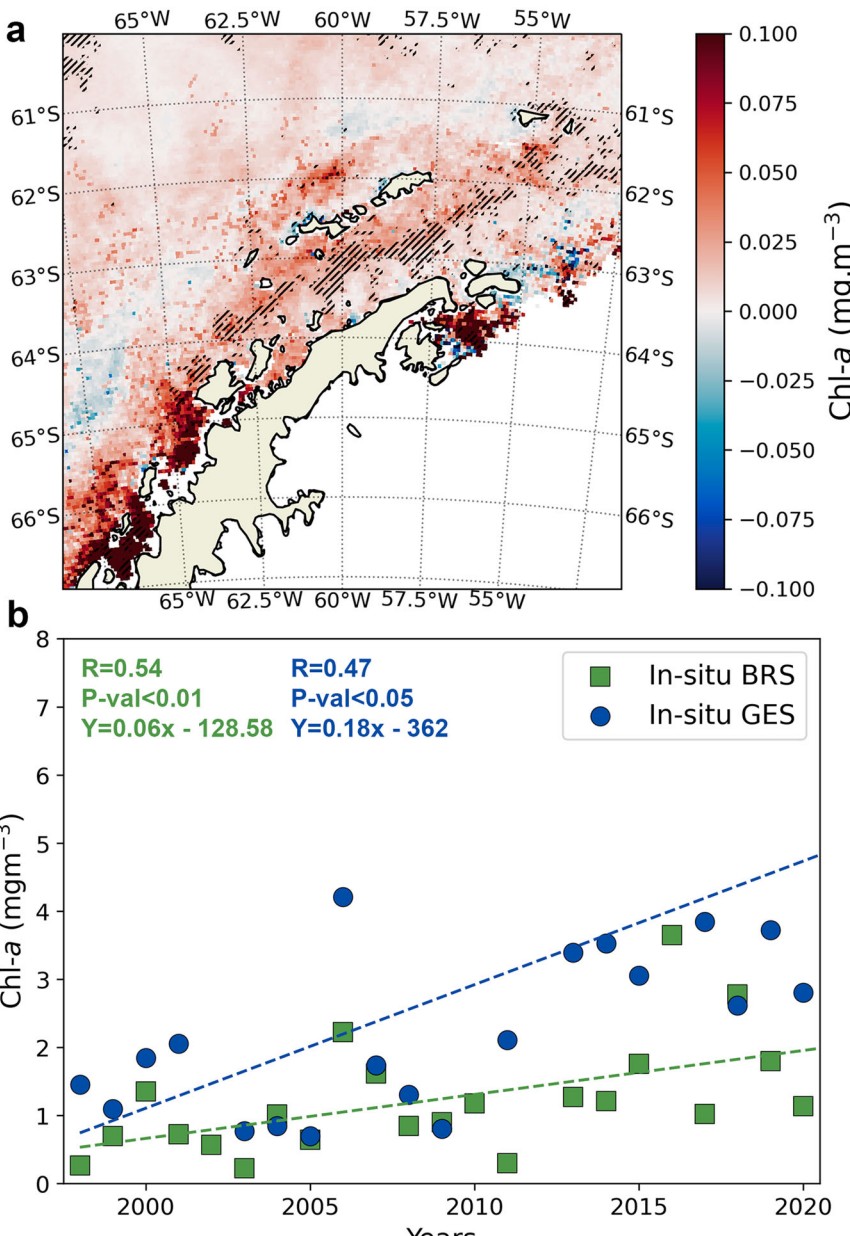

**Fig. 3 | Satellite and in-situ phytoplankton biomass increasing trends in the West Antarctic Peninsula. a** Calculated change in satellite-derived chlorophyll-*a* per year (mg m⁻³ year⁻¹) (September-April) along the Antarctic Peninsula between 1998-2022. Oblique black lines represent the areas with statistically significant changes (*p* value < 0.05; Spearman's rank correlation test). **b** Mean September-April in-situ chlorophyll-a (mg m⁻³) during the period 1998–2020 for the regions BRS (squares) and GES (circles). Dashed lines represent the statistically significant linear regression with a two-sided null hypothesis (*N* = 23; BRS: Pearson R = 0.58, *p* value = 0.0043; GES: Pearson R = 0.64, *p* value = 0.0026). Other regions were not included due to their low number of in-situ samples.

variability associated with bloom timing in this region, which might prevent the identification of linear trends within the 25 years of our dataset. Previous long-term studies on phytoplankton bloom phenology in the WAP are rare due to the scarcity of data that characterizes the region. Using +20-year data (1992-2016) from three different stations along the WAP, Kim et al[37]. showed that the northernmost station (Carlini Station; BRS region) exhibited a longer and less pronounced (i.e., with a lower biomass peak) bloom than the other stations (Palmer and Rothera)[37], in line with our results. However, these authors did not assess long-term changes in bloom timing. Recently, Thomalla et al[38]. suggested that there has been an overall tendency between 1998 and 2022 for blooms in the more heavily ice-influenced regions of the Southern Ocean to initiate earlier and have longer durations[38], which contradicts our results for the WAP. It is important to mention that

Thomalla et al. focused on the entire Southern Ocean, thus being less detailed at a regional scale, unlike the present study for the WAP. Another recently published study, focused on the WAP, also reported that phytoplankton blooms in the marginal ice zone and the continental shelf have shifted later[16], corroborating our results and reinforcing the importance of regional studies for the understanding of phytoplankton phenology in complex regions.

## Sea ice decline is enabling phytoplankton growth
The changes observed in phytoplankton biomass and bloom dynamics in spring and autumn can be mainly attributed to the long-term decline in sea ice coverage seen along the WAP over the past four decades, particularly during the late autumn – early winter transition[5–9,17]. A decrease in the mean sea ice extent was seen in both spring and

**Table 1 | Yearly change in satellite-derived phytoplankton biomass within each month along the Antarctic Peninsula during 1998–2022**

| Period evaluated | DRA | BRS | WED$_N$ | GES | WED$_S$ |
|---|---|---|---|---|---|
| September | **0.003\*\*** | **0.009\*\*\*** | 0.004 | 0.002 | **0.054\*** |
| October | **0.003\*** | **0.004\*** | 0.008 | 0.005 | 0.018 |
| November | 0.001 | 0.005 | 0.004 | 0.009 | −0.003 |
| December | 0.004 | 0.005 | 0.017 | **0.033\*** | 0.243 |
| January | 0.005 | 0.010 | −0.007 | 0.065 | 0.079 |
| February | 0.006 | 0.018 | 0.003 | 0.047 | 0.152 |
| March | **0.008\*\*** | **0.029\*\*\*** | **0.012\*\*** | **0.061\*\*\*** | **0.190\*** |
| April | **0.009\*\*\*** | **0.022\*\*\*** | **0.008\*** | - | - |
| Spring | 0.002 | **0.005\*\*** | 0.007 | 0.006 | 0.02 |
| Summer | 0.005 | 0.011 | 0.006 | 0.046 | 0.185 |
| Autumn | **0.008\*\*** | **0.011\*\*\*** | **0.012\*\*** | **0.058\*\*\*** | **0.189\*** |
| September–April | **0.004\*** | **0.027\*\*** | 0.008 | 0.033 | 0.122 |

For each region, the yearly linear change (mg m$^{-3}$ year$^{-1}$; slope of linear regression) in the spatially integrated mean chlorophyll-*a* is presented for each month between September and April, as well as for spring (September-November), summer (December-February), autumn (March–April) and full season (September–April). Bold *, **, and *** correspond to *p* value < 0.1, *p* value < 0.05, and *p* value < 0.01, respectively (a Pearson correlation test with a two-sided null hypothesis was used; see Supplementary Table 5 for a full list of the *p* values associated with each test).

**Table 2 | Yearly change in phytoplankton bloom phenology along the Antarctic Peninsula during 1998-2022**

| Phenology metric | DRA | BRS | GES |
|---|---|---|---|
| Bloom initiation | −0.003 | 0.058 | 0.008 |
| Bloom termination | **0.416\*\*\*** | 0.046 | **0.098\*\*** |
| Bloom peak | 0.097 | 0.115 | −0.021 |
| Bloom duration | **0.419\*\*\*** | −0.012 | **0.089\*** |

For each region, the yearly linear change (i.e., the slope of linear regression) was estimated for each bloom phenology metric calculated in this work: bloom initiation (weeks year$^{-1}$), bloom termination (weeks year$^{-1}$), bloom peak (weeks year$^{-1}$), bloom duration (weeks year$^{-1}$), and bloom magnitude (mg m$^{-3}$year$^{-1}$; a measure of the biomass accumulated during the bloom). Bold *, **, and *** correspond to *p* value < 0.1, *p* value < 0.05, and *p* value < 0.01, respectively (a Pearson correlation test with a two-sided null hypothesis was used; see Supplementary Table 5 for a full list of the *p* values associated with each test).

autumn periods from 1982–1997 to 1998–2022 for each region of the WAP (DRA, BRS, and GES; Fig. 4a; Supplementary Fig. 6). A key consequence of this decline is that sea ice retreat (in spring) and advance (in autumn) are occurring increasingly earlier and later, respectively, leading to an increase in the number of sea ice-free days (Fig. 4a). This allows for enhanced phytoplankton growth during spring and autumn in areas previously occupied by sea ice since the additional space and the improved light conditions are now more favourable for phytoplankton accumulation. The observed sea ice decline across the entire WAP likely explains why the phytoplankton growth in the autumn was widespread along the region.

On average, the autumn phytoplankton biomass in the WAP increased from 29% to 42% of the total biomass from September to April, while spring and summer biomass contribution slightly decreased (Fig. 4b). Several reasons can explain this difference. Firstly, the rate at which sea ice advance is becoming later is higher than the rate at which sea ice retreat is becoming earlier[5], i.e., the amount of the new ice-free days caused by the long-term warming is higher in the autumn than in the spring. Secondly, the advance of sea ice in the WAP is delayed by persistent northerly winds that occur during the autumn[5], meaning that sea ice coverage tends to be lower in the autumn than in the early spring. As such, there is more new space for phytoplankton to grow and accumulate biomass in the autumn than in the spring. Moreover, the lower sea ice coverage in the autumn may allow for higher light penetration in the upper layers of the water column[12], promoting phytoplankton growth in waters where biomass could not previously accumulate under the sea ice. Finally, phytoplankton biomass typically reaches much higher concentrations in the late summer compared to the early spring (Fig. 3a, b, d). Apart from the higher temperatures and lower sea ice coverage, a key reason for this is that the phytoplankton biomass already accumulated in the late summer can be maintained throughout the early autumn if enough nutrients are available, while phytoplankton in the early spring must grow from the typically very low winter biomass.

These findings put into perspective the future trends for sea ice extent in Antarctica. In the long term, sea ice is expected to decrease as warming continues[39], similar to what is also projected for the Arctic[40]. The resulting decrease in the duration of the sea ice season is likely to be followed by enhanced phytoplankton growth at the limits of the sea

ice season (spring and autumn). Nevertheless, even if all required nutrients for growth are available, the duration of phytoplankton blooms will ultimately be limited by the low light availability that characterizes the austral winter in Antarctica. In the Arctic, Ardyna et al.[41] showed that recent sea ice loss have triggered fall blooms analogous to what we observe in the WAP, with wind-driven vertical mixing playing an important role in supporting phytoplankton growth. More high-resolution in-situ studies focused on the austral spring and, particularly, autumn are essential for a better understanding of how phytoplankton dynamics will change. In addition, complementary sampling techniques to in-situ sampling expeditions such as multi-parametric (biogeochemical) floats and underwater gliders are becoming more frequent and are expected to contribute significantly to the acquisition of in-situ observations from other seasons besides summer[42–44].

## SAM as a key driver of regional variability
While the overall declining trend in sea ice explains most of the findings observed in our work, it does not explain some of the regional differences observed along the WAP. For instance, it does not explain why the more statistically significant pixel-wise biomass trends were found in the northernmost and more offshore areas of the WAP (Fig. 2a). This can be mainly attributed to an intensification in the Southern Annular Mode, a key climate mode modulating the marine environment of the WAP, particularly its northernmost sector. Under positive SAM conditions, circumpolar westerlies strengthen, bringing warmer air into the WAP and increasing cloud coverage[45]. In the marine ecosystems of the WAP, positive SAM conditions are typically associated with warmer surface temperatures and reduced sea ice[36]. Since the 1950s, the SAM index has shown a generally upward trend linked to anthropogenic climate change, although a plateau was observed between the late 1990s and the 2000s[36,45,46]. Since 2010, however, the SAM index has intensified again and is becoming increasingly positive at a fast rate, as seen by the large increase in the cumulative SAM index (Fig. 5a).

We hypothesize that this recent intensification of SAM may have contributed to drive the observed region increase in phytoplankton biomass as well as the reason why this was more prevalent in the northwestern Peninsula. We observed a positive and significant pixel-wise correlation (considering the average between September-April period) between the average Chl-*a* concentration and SAM index in the waters along most of the Western Antarctic Peninsula, particularly for the southern Drake Passage and the northern tip of the Antarctic Peninsula (Fig. 5b). As previously mentioned, wind intensification associated with positive SAM results in deeper mixed layers along the Antarctic Peninsula, especially in offshore waters[38]. The increased vertical mixing caused by stronger winds could be increasing micro-nutrient availability, particularly iron (Fe), for the DRA and the non-sheltered areas of the BRS, which are less affected by sea ice conditions and have a stronger oceanic influence. This would help prevent

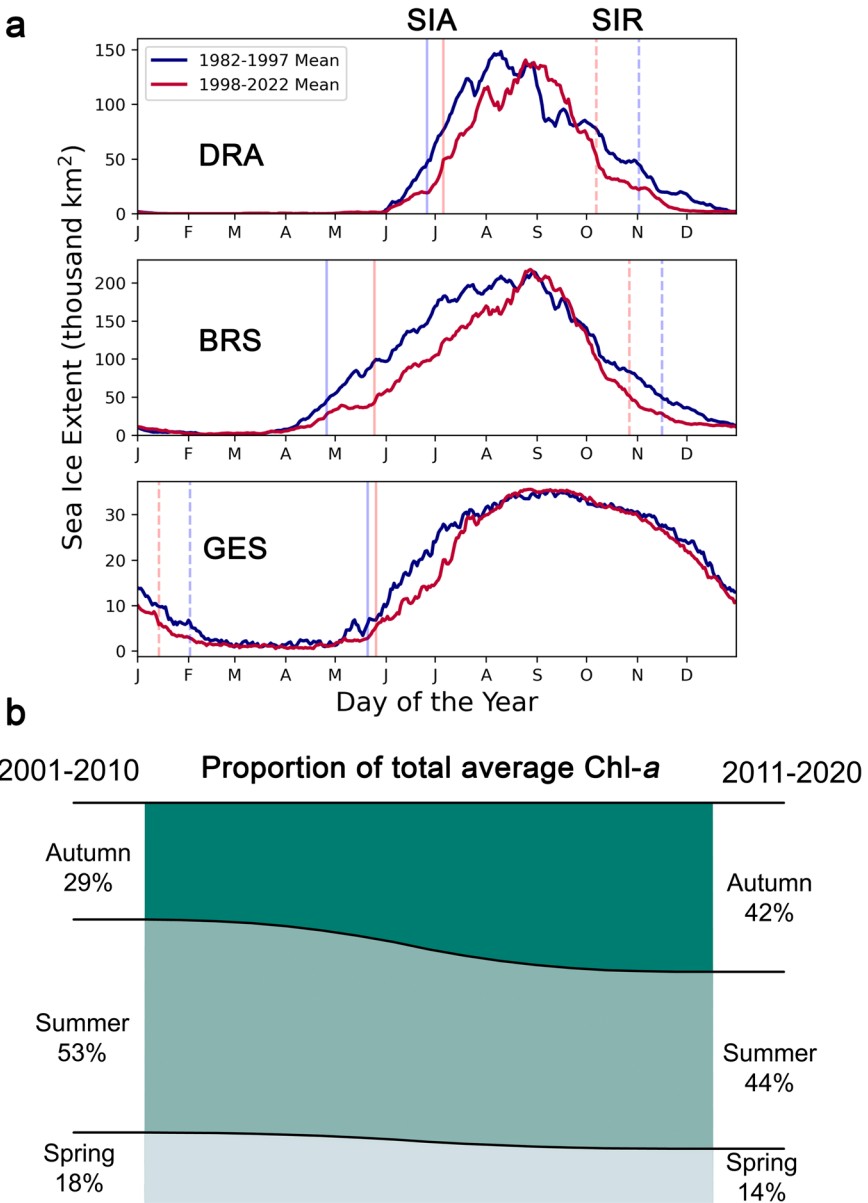

**Fig. 4 | Changes in sea ice season timing and effects on the proportion of biomass in the WAP attributed to each season. a** Yearly average cycle of the Sea Ice Extent (thousand km²) in each region (DRA, BRS, GES) between 1982–1997 (blue) and 1998–2022 (red). For each period, vertical lines indicate the average date of sea ice advance (SIA; solid) and sea ice retreat (SIR; dashed). **b** Change in the average proportion of total chlorophyll-*a* (Chl-*a*) concentration attributed to each season from 2001–2010 to 2011–2020 in the WAP (regions DRA, BRS and GES). Spring: September-November, Summer: December-February, Autumn: March-April.

nutrient limitation in these areas, especially in the autumn, when a greater amount of nutrients has already been consumed[27,28], coinciding with the period when most of the biomass enhancement was observed in our study (Table 1). Accordingly, Tagliabue et al.[47] observed that Fe supply from winter mixing is crucial to offshore primary production in the Southern Ocean. Furthermore, the stronger westerlies associated with positive spring SAM can also enhance warm, nutrient-rich CDW intrusions over the shelf during summer[36,48,49]. Such intrusions propagate along the shelf, eventually reaching the northern Antarctic Peninsula and its more offshore areas[36]. Several studies have already shown high phytoplankton biomass linked to CDW intrusions[48,49], suggesting that these ocean circulation processes coupled with positive SAM trends could be another contributing factor to the increased biomass in these regions.

Contrastingly, we do not observe a positive correlation between SAM and Chl-*a* in the southernmost coastal WAP (GES; Fig. 5b).

Enhanced CDW intrusions have been reported to force and increase glacier retreat due to its warmer temperatures, leading to more frequent and stabilized water column structures during spring and summer months[15]. While such changes in the upper ocean physical compartments may have initially led to an increase in summer biomass[15], the recent intensification of SAM could be driving shifts in phytoplankton community composition and size structure and, consequently, on biomass accumulation. Recent studies along the WAP have reported frequent shifts from large centric diatoms to smaller-sized cryptophytes in regions under localized glacier meltwater input during the austral summer[13,18,50–52]. As a result, this shift could decrease the total summer phytoplankton biomass accumulation under positive SAM periods, explaining the slightly negative correlation between SAM and Chl-*a* for the coastal regions observed here. Moreover, it will also be essential to consider the influence the El-Niño Southern Oscillation can have on the interannual dynamics of phytoplankton, as summers

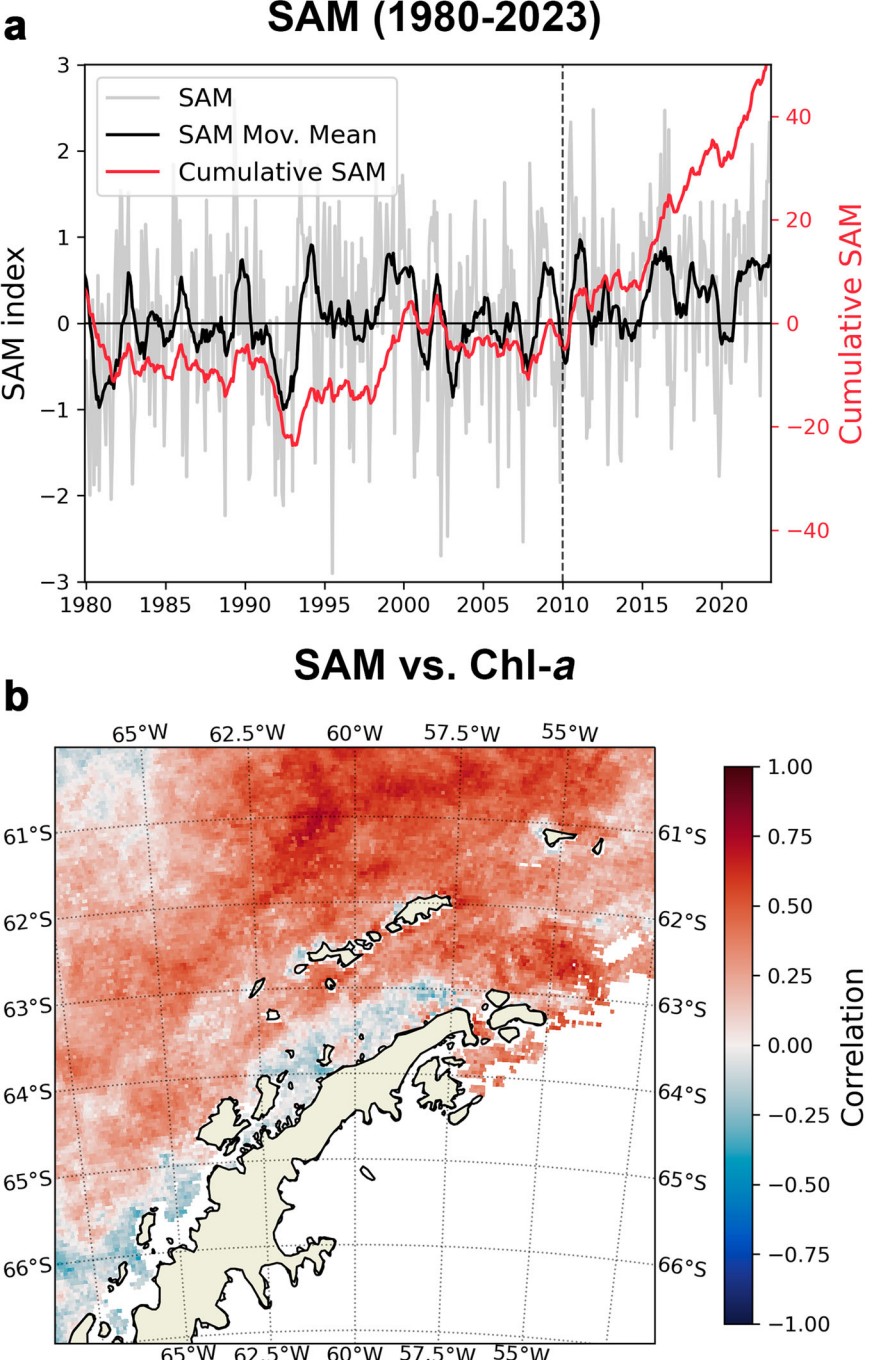

**Fig. 5 | Recent changes in the Southern Annular Mode and its correlation with phytoplankton biomass along the Western Antarctic Peninsula. a** Monthly-averaged Southern Annular Mode index (SAM; grey), and its 1-year moving mean (black) and cumulative sum (red, secondary axis) since 1980. A vertical dotted line indicates 2010. **b** Spatial correlation between the mean SAM indexes and chlorophyll-*a* (Chl-*a*) between September-April during 1998–2022. Pixels with statistically significant linear trends are indicated.

following extreme El-Niño events have already been associated with large diatom blooms[53,54].

A preferential feeder on large diatoms[55–57], Antarctic krill (*Euphasia superba*) is a key species in the Southern Ocean ecosystem and an essential link between phytoplankton and higher trophic levels such as whales, seals, and penguins[4,11,12]. Therefore, localized shifts from diatoms to cryptophytes could likely contribute to shaping the phytoplankton dynamics along the coastal WAP by alleviating the grazing pressure from the krill on phytoplankton biomass accumulation, while also potentially impacting the krill stocks. Nevertheless, it is still

unclear what impacts such changes in phytoplankton community composition and size structure may have on the WAP marine ecosystems. Further in situ studies are required to understand if a phytoplankton community shift to cryptophytes will significantly weaken biological carbon uptake or lead to structural changes in the WAP coastal marine food web.

## Methods

A summary of all datasets used, including their temporal coverage, spatial and temporal resolution, units, product name and relevant

references is available in the Supplementary Table 3. The following subsections present further detail on each dataset and presents all methodologies used. All analysis were performed in Python 3.8.8.

## Satellite data

Daily satellite remote sensing reflectance data with 4 km spatial resolution was extracted from the ESA Ocean Colour Climate Change Initiative (OC-CCI) v6.0 product[26] for 1998-2022. Chlorophyll-*a* (mg m$^{-3}$) was estimated by applying the OC4-SO algorithm[22], a regional Chlorophyll-*a* algorithm specifically devised for the Antarctic Peninsula region to increase the accuracy of Chlorophyll-*a* measurements[22]. Satellite sea surface temperature (°C) and sea ice concentration (%) fields with daily temporal resolution and ~5 km spatial resolution were acquired from the Operational Sea Surface Temperature and Ice Analysis (OSTIA)[58], available at the Copernicus Marine Environment Monitoring Service (https://marine.copernicus.eu/). The sea ice concentration was used to calculate daily sea ice extent (i.e., the area occupied by sea ice) for the clusters DRA, BRS and GES by multiplying the area of each pixel (~25 km$^2$) by the number of pixels covered by sea ice (using a minimum threshold of 15% of sea ice concentration per pixel[5]). For each cluster, the annual sea ice duration, day of sea ice advance, and day of sea ice retreat were calculated following Stammerjohn et al[5]. Following this definition, the day of sea ice advance corresponded to the first day in which sea ice concentration is higher than 15% for at least 5 consecutive days. The day of sea ice retreat is the inverse, i.e., the first day after the winter when the sea ice concentration drops below 15% (a 5 consecutive day window is also used)[5].

Daily, 4 km resolution photosynthetically active radiation (PAR; Einstein m$^{-2}$ d$^{-1}$) data were retrieved from the ESA GlobColour project[59]. Monthly Southern Annular Mode (SAM) index values were retrieved from the NOAA Centre for Weather and Climate Prediction Centre (https://www.cpc.ncep.noaa.gov/).

## In-situ data

HPLC-derived in-situ surface measurements of Chl-*a* collected across the Western Antarctic Peninsula were used. The full dataset ($N = 4322$) included published data available in Valente et al.[60] and & Palmer LTER[61], as well as unpublished data collected by the Brazilian High-Latitude Oceanography Group (GOAL-FURG).

The GOAL-FURG in-situ Chl-*a* measurements were collected during 12 austral summer expeditions to the WAP aboard the Brazilian vessels NP Almirante Maximiano and NP Ary Rongel ($N = 533$). Seawater samples (ranging from 0.5 to 2.5 L) were filtered under low vacuum using GF/F filters (25 mm diameter, 0.7 μm pore size) and subsequently stored at −80 °C. Prior to analysis, the filters were placed in screw-cap centrifuge tubes with 3 mL of 95% methanol (2% ammonium acetate) containing 0.05 mg L−1 trans-β-apo-8'-carotenal as an internal standard. Samples were sonicated for 5 min, stored at −20 °C for one hour and then centrifuged at 1100 × *g* for 5 min at 3 °C. The supernatants were filtered through PTFE membrane filters (0.2 μm pore size) to ensure no residues were inserted in the HPLC system. For pigment analysis, a monomeric C8 column and a mobile phase containing pyridine was used. Before injection, 1 mL of sample was mixed with 0.4 mL of Milli-Q water in 2.0 mL glass sample vials and placed in the HPLC cooling rack of a Shimadzu Prominence LC-20A Modular HPLC System. The method's detection and quantification limits have been calculated and discussed by Mendes et al[62]. Pigment standards from DHI (Institute for Water and Environment, Denmark) were used for calibration of the HPLC system. Pigments were identified based on their absorbance spectra and retention times from the signals in the photodiode array detector (SPD-M20A; 190–800 nm; 1 nm wavelength accuracy) or fluorescence detector (RF-10AXL; Ex. 430 nm/Em.670 nm). Peaks were integrated using the LC-Solution software, but all peak integrations were manually verified and corrected if necessary. A quality assurance threshold procedure, through the application of limits of quantification and detection, was applied to the pigment data to reduce the uncertainty of pigments found in low concentrations[63]. Pigment concentrations were normalized to the internal standard to account for losses and volume changes.

Combined, the three databases result in a comprehensive dataset of in situ Chl-*a* measurements spanning from 1998 to 2020. In-situ atmospheric wind speed measured at Palmer Station were extracted from the 'Palmer Station Weather−Daily Averages (#28)' dataset, available in EDI[64]. In-situ atmospheric wind speed measured at Marambio and Great Wall stations were extracted from the Global Historical Climatology Network daily (GHCNd)[65].

## Clustering areas with coherent phytoplankton seasonal patterns

Due to the extent and complexity of the marine environments off the Antarctic Peninsula, an agglomerative hierarchical clustering analysis was first used to cluster areas within the Antarctic Peninsula with coherent phytoplankton seasonal patterns using remote sensing data from 1998 to 2022. First, for each pixel, the following metrics were calculated: (i) the first month when phytoplankton growth (i.e., satellite Chl-*a*) is, on average, detected after the winter, (ii) the last month when phytoplankton growth is, on average, detected after the summer, (iii) the average month when the peak satellite Chl-*a* is detected, and (iv) the numerical integration of mean Chl-*a* concentrations during the austral summer (November-February), calculated using Simpson's rule. All metrics were normalized. Finally, the spatial resolution of the pixels was reduced to 10 km to avoid performance issues when running the clustering analysis. The distance and linkage metrics used for clustering were the Manhattan distance and average linkage, respectively. Four clusters were initially identified. After closer inspection, one cluster was divided in two due to a partial influence of sea ice, which could affect phytoplankton bloom phenology, thus bringing the total number of clusters to 5. All pixels that were, on average, covered with sea ice during the entire year (i.e., sea ice concentration above 15%) were discarded. Subsequently, pixels indicative of noise (e.g., pixels of a cluster located inside another cluster without a logical reason) were manually removed. Finally, the spatial resolution of the resulting map of clusters was upscaled to 4 km. All analyses were performed using Scipy and Sklearn modules in Python 3.8.8.

## Phytoplankton bloom phenology

For each of the five identified clusters, phytoplankton bloom phenology metrics were derived from satellite Chl-*a*. First, daily satellite Chl-*a* was temporally averaged to an 8-day week frequency and spatially averaged within each cluster. Subsequently, the phytoplankton austral spring-summer bloom was identified as the event, no shorter than 2 weeks, in which Chl-*a* surpasses a threshold of 5% of the annual Chl-*a* median[66–68]. Five bloom phenology metrics were then calculated. The bloom initiation and bloom termination dates were defined as the first and last week of the bloom, respectively (i.e., when Chl-*a* rises above and subsequently falls below the 5% threshold). Bloom duration corresponds to the number of weeks between bloom initiation and bloom termination. The bloom peak date is the week within the blooming period when the maximum Chl-*a* value is found.

## Analysis of phytoplankton trends

To understand if phytoplankton biomass and bloom phenology metrics in the Western Antarctic Peninsula have changed since 1998, the time series for each previously defined cluster were assessed for statistically significant trends or environmental shifts. For each year, the mean Chl-*a* during each month between September and April (full austral spring-summer period) was calculated, as well as the mean Chl-*a* during this entire period (September-April) and during only the peak of the summer (December-February). The regions within the eastern sector of the Antarctic Peninsula (WEDi and WEDo) were excluded from these

analyses since satellite observations are very limited due to the persistence of sea ice that characterizes this sector. Since low data availability and sea ice adjacency is likely to affect the validity of trend analyses. Linear trends were estimated by fitting linear regressions to each time series and extracting the slope of the resulting line. Outliers were excluded by removing all Chl-$a$ values over (under) the mean Chl-$a$ + (−) 3 standard deviations. Pixel-wise linear regressions were also performed for the mean Chl-$a$ during September-April in order to produce maps showcasing the linear trend for Chl-$a$. In order to understand if a similar trend was observed in in-situ data, the mean Chl-$a$ during September-April was also calculated using only in-situ data from the GES and BRS regions (the only regions of the WAP where data was abundant enough to accurately run trend analyses). To understand how the trends in phytoplankton biomass may have changed between decades in the WAP, the average Chl-$a$ concentration for each season (spring, summer, and autumn) was calculated for 2001–2010 and for 2011–2020. Data from the three regions of the WAP (DRA, BRS and GES) were used for this comparison. The proportion of the Chl-$a$ attributed to each season to the total Chl-$a$ (the sum of the average spring, summer, and autumn mean Chl-$a$) was then calculated. Finally, to better understand the relationship between Chl-$a$ and SAM, a pixel-wise linear regression was performed for the full WAP region.

### Reporting summary

Further information on research design is available in the Nature Portfolio Reporting Summary linked to this article.

## Data availability

The previously unpublished in-situ chlorophyll-a dataset from GOAL utilized in this study is now accessible at https://doi.org/10.5281/zenodo.12580624. All other data utilized in this study have been previously published and are available online. The OC-CCI satellite chlorophyll-a product is available at https://www.oceancolour.org/. The in-situ chlorophyll-a data acquired from Valente et al[60]. are available at https://doi.org/10.5194/essd-14-5737-2022. The in-situ chlorophyll-a data acquired from Palmer-LTER are available https://doi.org/10.6073/pasta/9d0c1561ca8c5540227df6efa37c61b5. The OSTIA product, containing sea surface temperature and sea ice concentration data is available at https://doi.org/10.48670/moi-00168. The GlobColour PAR data is available at https://hermes.acri.fr/. The wind data from ERA5 Reanalysis are available at https://doi.org/10.24381/cds.adbb2d47. The in-situ wind data from Palmer Station are available at https://doi.org/10.6073/pasta/3eefb45dbfb784c3cabe3690ea46fe9e. The in-situ wind datasets from Marambio Station and Great Wall Station are available at https://www.ncei.noaa.gov/products/land-based-station/global-historical-climatology-network-daily. The Southern Annular Mode (SAM) data is available at https://www.cpc.ncep.noaa.gov/. Source data are provided with this paper.

## Code availability

All custom code developed within this work is available at: https://github.com/afonsomferreira/antarcticpeninsula-trends.

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

## Acknowledgements

A. Ferreira received a Ph.D. grant (SFRH/BD/144586/2019; https://doi.org/10.54499/SFRH/BD/144586/2019) from Fundação para a Ciência e a Tecnologia (FCT). A.C. Brito (CEECIND/00095/2017) and C.V. Guerreiro (CEECIND/00752/2018/C, doi: 10.54499/CEECIND/00752/2018/CP1534/CT0011) were supported by the Scientific Employment Stimulus Programme, also funded by FCT. C.V. Guerreiro also benefited from a Marie Sklodowska-Curie European Fellowship supported by the EU H2020-MSCA-IF-2017 (grant no. 796802) linked to project DUSTCO. C.R.B. Mendes (PQ 312569/2021-1) and E.R. Secchi (PQ 310597/2018-8) were both granted with a researcher fellowship from National Council for Research and Development (CNPq). A MSc fellowship from CAPES was granted to R.R. Costa. This work was also funded by the European Union's Horizon 2020 Research and Innovation Programme under grant agreement N 810139: Project Portugal Twinning for Innovation and Excellence in Marine Science and Earth Observation—PORTWIMS. This is a multidisciplinary study as part of the Brazilian High Latitude Oceanography Group (GOAL) activities in the Brazilian Antarctic Programme (PROANTAR). Financial support was also provided by CNPq and Coordination for the Improvement of Higher Education Personnel (CAPES). This study was conducted within the activities of the PROVOCCAR and ECOPELAGOS projects (CNPq grant nos. 442628/2018-8 and 442637/2018-7, respectively), and is within the scope of two Projects of the Institutional Internationalization Programme (CAPES PrInt-FURG – Call no. 41/2017). CAPES also provided free access to many relevant journals through the portal "Periódicos CAPES". This study had the support of FCT through the strategic projects UIDP/04292/2020 (https://doi.org/10.54499/UIDP/04292/2020) and UIDB/04292/2020 (10.54499/UIDB/04292/2020) awarded to MARE and through project LA/P/0069/2020 (https://doi.org/10.54499/LA/P/0069/2020) granted to the Associate Laboratory ARNET. It also received support from FCT through the Portuguese Polar Programme (PROPOLAR) calls in 2018/2019, 2019/2020, and 2020/2021. This research is framed within the College on Polar and Extreme Environments (Polar2E) of the University of Lisbon. Authors are also indebted to the ESA Ocean Colour—Climate Change Initiative project (https://esa-oceancolour-cci.org/), the Copernicus Marine Environment Monitoring Service (https://marine.copernicus.eu/), the GlobColour project (https://www.globcolour.info/), the NOAA Centre for Weather and Climate Prediction Centre (https://www.cpc.ncep. noaa.gov/), the NOAA Physical Sciences Laboratory (https://psl.noaa.gov/) and the Palmer Station Antarctica Long-Term Ecological Research (https://pallter.marine.rutgers.edu/) for providing free access to the datasets used in this work.

## Author contributions

A.F., C.R.B.M., A.C.B., and V.B. conceptualized the study. C.R.B.M., A.C.B., and V.B supervised the project. A.F., C.R.B.M, A.C.B, R.R.C designed the methodology. A.F., C.R.B.M, R.R.C., V.M.T., and C.V.G. were responsible for the in-situ data collected within this work. A.F. performed the remote sensing and statistical analyses. A.F. wrote the original draft. C.R.B.M, R.R.C., V.B., V.M.T., C.V.G., E.S., and A.C.B. revised and edited the manuscript. C.R.B.M., A.C.B., V.B, and E.S. were responsible for funding acquisition. All authors have read and agreed to the published version of the manuscript.

## Competing interests

The authors declare no competing interests.
