## [Peer Review File · Nature Communications]

Climate change is associated with higher phytoplankton biomass and longer blooms in the West Antarctic PeninsulaEditorial note: Parts of this Peer Review File have been redacted as indicated to remove third-party material where no permission to publish could be obtained.

REVIEWER COMMENTS

Reviewer #1 (Remarks to the Author):

Review of “Phytoplankton trends in a changing climate: higher biomass and longer blooms in the West Antarctic Peninsula” submitted to Nature Communications by Ferreira et al.

In this paper, the authors look at 25 years of satellite derived ocean color in the waters of the west Antarctic Peninsula to understand the impact of climate change on the phytoplankton phenology and biomass. They find an interesting result in that the changes in the sea ice cover linked to the impacts of climate change have mainly led to an increase in fall phytoplankton blooms, something that is new to the Southern Ocean. This is an interesting study and could be worth considering in Nature Communications if some of the caveats I detail below are corrected.

My first major comment is that it is not clear enough throughout the manuscript what period is used for the analyses. In the abstract, the authors state they ran their analyses from 1998 to 2022, but the final year 2021 is mentioned in several places in the manuscript, figures, and supplementary figures. The authors need to give precisely what years and months they have used for their analyses. See my further comments below.

For my second major comment, I believe that the referencing in the present paper is not fully correct and leads to some flawed argumentation around the interpretation of the authors’ results. I explain why I think this is a problem in detail here. In the introduction and throughout their paper, the authors refer to previous seminal papers that have studied a relatively similar subject, climate change in the waters west of the Antarctic Peninsula and its impacts on the marine ecosystems. However, these studies are almost all 10 or more years old. And the climatic trends that these papers presented seemed to have changed. In fact, the most conspicuous climatic trend until 2010 was an earlier sea ice retreat and a later

ice advance as reported in, for example, Stammerjohn et al. 2008. However, the trend has changed, and the sea ice retreat has not particularly become earlier since 2010 (see the authors' Figure 4a for example). Therefore, the authors' reasoning regarding the role of the sea ice retreat and advance cannot be fully verified at this stage. I would suggest that the authors present the sea ice phenological changes focused on the period they study in their paper (1998-2022) and not for the entire time series available for satellite derived sea ice data (starting in 1978 or 1980 as chosen by the authors). Only by doing this, will the authors be able to study the possible role of changes in sea ice phenology on the observed changes in phytoplankton biomass and phenology. The main result of this paper, the change in phytoplankton biomass that the authors mainly observe in autumn is an interesting result, but as of now, the supporting data are not well enough articulated. The authors should also acknowledge throughout their paper that the time series they study is different than the ones previously studied for this region, explaining the different results obtained.

In parenthesis, this reminds me very much of an Arctic Ocean paper that found similar phenological changes following the well-known sea ice changes that have happened in the Arctic Ocean, in case the authors did not know of it: Ardyna, M., M. Babin, M. Gosselin, E. Devred, L. Rainville, and J.-É. Tremblay (2014), Recent Arctic Ocean sea ice loss triggers novel fall phytoplankton blooms, *Geophys. Res. Lett.*, 41, 6207–6212, doi:10.1002/2014GL061047

At the moment, I suggest the authors to consider my suggestions and those of other reviewers, to improve the quality of their study for consideration of publication in *Nature Communications*.

I wish the authors the best of luck with the review process.

Best regards

Specific comments:

Line 20: in this place, the authors mention that they are going to present their analyses over 25 years of data (1998-2022), but the final year of analyses is presented as 2021 in many

other places of the manuscript. So I would suggest the authors to be very precise and also include the month for the start and end dates, since we are dealing here with Austral primary productivity seasons.

Line 27: the end year for the “recent intensification of the Southern Annular Mode” is not given in this sentence.

Lines 29-30: “region” is repeated twice in this sentence. Please remove one occurrence.

Line 30: “raise important critical questions”. Please choose between “important” and “critical”, no need to use both adjectives in this sentence.

Lines 34-38: In the first lines of the introduction, where the references #1, #2, #4 and #5 are used to describe the environmental changes that have happened in the Peninsula over the last decades, the authors must bring more recent references of the physical changes that have occurred in the WAP since the most recent of these references (i.e. Montes-Hugo et al. 2009). I will come back to this question later, but I think it is essential to follow the argumentation of the authors that we also know what physical changes have happened in the WAP since these seminal studies were published.

Line 85: when mentioning Figure 2 here, there also seems to be inconsistencies regarding the dates which are given as “1998-2022” in the Figures panels and as “1998-2021” in the Figure legend (line 450). Please correct throughout the manuscript.

Lines 86-90: so does that mean that the other regions are less likely to experience iron limitation? The authors should comment on this in the paper and bring previous references.

Line 108: here the period stated is 1997-2021, contrary to what is indicated in the abstract. In the legend of Figure 3 (line 458), the period indicated is 1997-2022. And in the legend of Supplementary Figure 3 (line 16), the period indicated is also 1997-2022. The authors should be consistent throughout the paper regarding the exact period they ran their analyses on.

Line 119: “2011-2011” should be “2011-2021” I believe. However, the authors should make sure the period studied is consistent throughout the paper.

Lines 134-138: the authors’ Supplementary Figure 3b also suggests this increase in phytoplankton biomass in the Gerlache Strait in summer.

Lines 150-151: this paragraph by the authors is about the increase in phytoplankton biomass in the northern WAP (BRS and DRA). However, this sentence and the Supplementary Figure 5 present data from the Southern (or mid-) WAP (i.e. Palmer Station). Therefore, this argumentation is not valid and the authors should present wind data trends for the whole WAP to feed their argumentation.

Lines 144-157: I think that the major problem in this paragraph and in other places of this paper is that the authors compare their long-term study (1998-2022) with other long-term studies that were performed using data from different periods. For example, the Montes-Hugo et al. (2009) study compares two very different decades (1978-1986 and 1998-2006) than the present paper. However, the physical dynamics of the WAP have not evolved linearly during the period presented in Montes-Hugo and the period presented in this paper. This is the case of sea ice phenology for example. Therefore, the present study is not comparable to these previous seminal studies and the authors should be more cautious when comparing their results with these previous studies.

Line 185: in the legend of Figure 4a, only the equation for the linear regressions for the sea ice advance is given. The authors should also give the equation for the linear regressions for the sea ice retreat which they also plot.

In Figure 4, I think that this would be important to highlight the trends that correspond to the period that the authors use to study the change in phytoplankton biomass and phenology, so 1998-2022. This will be important when trying to interpret the role of sea ice phenology changes in phytoplankton dynamics. What happened in the 2 decades prior to 1998 is interesting but not relevant regarding the data presented in this paper.

Line 187: I guess that the authors should also explain somewhere in the manuscript, perhaps here, that sea ice presence acts to decrease light penetration to the upper ocean. Even though relatively known, this may still be useful to some readers.

Line 191: The statement that sea ice declining rates are higher in the Gerlache Strait compared to the Drake Passage and Bransfield Strait needs to be backed up by showing relevant sea ice trends in the aforementioned regions or by presenting a reference that has shown this previously, if possible, a recent reference.

Line 193: “while spring and summer biomass contribution slightly decreased...” add missing word.

Line 194-196: since the authors only give the equation of the linear regression for sea ice advance, the reader cannot judge if this statement is true. Again, here, only the period 1998-2022 will be relevant to the present study.

Line 201-204: while I agree with the authors’ statement that phytoplankton stocks are important here, I think that the authors should still consider and mention nutrients’ limitation for autumnal primary productivity.

Line 204-206: as of now, this statement cannot be verified since the authors do not show the trend regarding sea ice retreat between 1998 and 2022, and instead use a much longer time series starting in 1980. This statement also contradicts the information the authors give in their Supplementary Figure 2.

Line 238: this is not entirely clear to me if the authors tested the correlation between the annual(?) SAM index and the annual(?) average Chl-a concentration for each pixel of the region, or they used an interannual(?) mean. I would think that the first is true but perhaps the authors could clarify.

Line 269: Should this sentence read “following strong El-Nino...”?

Supplementary Figures:

Line 5: why is the period used for these data (1997-2021) different than the period used for the overall study (1998-2022)?

Line 10: why is the period used for these data (1997-2021) different than the period used for the overall study (1998-2022)?

Reviewer #2 (Remarks to the Author):

The paper “ Phytoplankton trends in a changing climate: higher biomass and longer blooms in the West Antarctic Peninsula” by Ferreira and colleagues describes a satellite-based study of the changes in phytoplankton biomass and phenology in waters around the Antarctic Peninsula. Although the title refers to the west side of the peninsula, the east side was considered as well. The paper provides a nice description of the changes over the last few decades and provides reasonable explanations for them. I found the paper to be informative and interesting and their methods robust. I only have a few minor issues that the authors should consider when revising their manuscript. I describe them below.

Line 22. What is meant by climate change here? Is this referring to anthropogenic or natural climate changes? This should be made clear here.

Line 26-27. Do the authors claim it is anthropogenic climate change or the SAM that is driving phytoplankton phenology? Or do they believe that anthropogenic climate change is driving the SAM? They need to clarify if these changes are part of a natural climate cycle or something induced by human activities.

Line 90-91. What differences are being referred to here? The authors have only described one subregion at this point.

Line 114. These numbers need more specific units. I assume that this is chlorophyll, but it needs to be specified.

Line 127-128. Could these observations be related to later ice freeze-up?

Line 137-138. Is this increase statistically significant? If the cutoff for significance is 0.05, then it is not statistically significant, and the authors should not say that there was an increasing trend.

Line 152-153. Has a long-term deepening of the mixed layer been observed, or is this only assumed from the stronger winds?

Line 188-189. I'm not sure what generalized means. Is it that phytoplankton growth was the same throughout the WAP?

Line 209-211. Do nutrients not play a role at all? Eventually, the iron will be consumed and the bloom will end, unless there are mechanisms to mix new iron to the surface. A longer growing season will not necessarily guarantee higher production. Plus, it is new production that is important to the ecosystem. Continuing a bloom for longer on recycled nutrients provides fewer benefits for the ecosystem and does not sequester carbon.

Line 260. Maybe change "compartments" to "components" ?

Line 268-270. If it is important to consider ENSO, then why hasn't it been considered in this study?

Line 272-273. Wouldn't the bigger issue be the decline in krill and consequently the animals that rely on them for food?

Line 510. Insert space (5 km)

Line 512. Change "photosynthetic" to "photosynthetically"

Line 513. Should be "data were"

Line 518. What is the sea ice concentration cutoff used to separate the sea ice season from the open water season? It may be described later, but should be given here first.

Line 542. Insert space (10 km)

Line 551. Insert space (4 km)

Point by point response to reviewers

All comments by Reviewers are presented in black.

All responses by the Authors are present in purple.

Please note that all line numbers included in this response correspond to the manuscript with tracked changes.

Reviewer #1:

Review of “Phytoplankton trends in a changing climate: higher biomass and longer blooms in the West Antarctic Peninsula” submitted to Nature Communications by Ferreira et al.

In this paper, the authors look at 25 years of satellite derived ocean color in the waters of the west Antarctic Peninsula to understand the impact of climate change on the phytoplankton phenology and biomass. They find an interesting result in that the changes in the sea ice cover linked to the impacts of climate change have mainly led to an increase in fall phytoplankton blooms, something that is new to the Southern Ocean. This is an interesting study and could be worth considering in Nature Communications if some of the caveats I detail below are corrected.

Authors: We are grateful to Reviewer #1 for taking the time to review the manuscript. We will now respond to each major comment sequentially.

My first major comment is that it is not clear enough throughout the manuscript what period is used for the analyses. In the abstract, the authors state they ran their analyses from 1998 to 2022, but the final year 2021 is mentioned in several places in the manuscript, figures, and supplementary figures. The authors need to give precisely what years and months they have used for their analyses. See my further comments below.

Authors: We acknowledge that the references to the temporal extent of the datasets used in the study were not sufficiently clarified in the original submitted version of the manuscript. This was primarily due to the use of multiple datasets from various sources, some of which did not originally encompass data up to 2022 at the time of the initial analyses. For instance, although the used OC-CCI Chl-a dataset already spanned from 1998 to 2022, certain environmental datasets (SST, Sea Ice concentration, and PAR) only contained data up to December 2021 when analysed.

This issue has been addressed in the new and revised version of the manuscript, for which we have:

1. Updated the satellite SST, Sea Ice, and PAR datasets to now also include data up to December 2022 and subsequently re-ran the analyses with the updated datasets. It should be highlighted that, not only this did not alter our originally submitted findings (refer to the updated Supplementary Table 2), but has provided even more consistency to our study.
2. Ensured that all references to dataset periods in the text and figures accurately reflect the timeframe of 1998-2022 and, when not, clearly indicating it in the text and figures (e.g., lines 110, 503, 510, 518, 544).

3. Introduced a new Supplementary Table that succinctly lists each dataset utilized, including their temporal coverage, spatial and temporal resolution, units, product name and relevant references (new Supplementary Table 3).

For my second major comment, I believe that the referencing in the present paper is not fully correct and leads to some flawed argumentation around the interpretation of the authors' results. I explain why I think this is a problem in detail here. In the introduction and throughout their paper, the authors refer to previous seminal papers that have studied a relatively similar subject, climate change in the waters west of the Antarctic Peninsula and its impacts on the marine ecosystems. However, these studies are almost all 10 or more years old. And the climatic trends that these papers presented seemed to have changed. In fact, the most conspicuous climatic trend until 2010 was an earlier sea ice retreat and a later ice advance as reported in, for example, Stammerjohn et al. 2008. However, the trend has changed, and the sea ice retreat has not particularly become earlier since 2010 (see the authors' Figure 4a for example). Therefore, the authors' reasoning regarding the role of the sea ice retreat and advance cannot be fully verified at this stage. I would suggest that the authors present the sea ice phenological changes focused on the period they study in their paper (1998-2022) and not for the entire time series available for satellite derived sea ice data (starting in 1978 or 1980 as chosen by the authors). Only by doing this, will the authors be able to study the possible role of changes in sea ice phenology on the observed changes in phytoplankton biomass and phenology. The main result of this paper, the change in phytoplankton biomass that the authors mainly observe in autumn is an interesting result, but as of now, the supporting data are not well enough articulated. The authors should also acknowledge throughout their paper that the time series they study is different than the ones previously studied for this region, explaining the different results obtained.

Authors: We acknowledge Reviewer #1's concerns and provide a thorough feedback to his/her comments below.

Indeed, the discussion of our manuscript revolves around key seminal papers, several of which published prior to 2010. We chose to emphasize these studies because they have crucially contributed to the current understanding of climate change impacts on the ecosystems of the Western Antarctic Peninsula (WAP). Nevertheless, we understand how important it is that the climate trends previously seen for the WAP remain true after 2010. We also agree with the reviewer that the original Figure 4a does appear to hint that the long-term trends in sea ice advance and retreat may have plateaued or even reverted recently, although no statistical significance was found after testing. Due to the important role sea ice can have on phytoplankton growth, particularly in the spring and autumn, we have redone the analysis.

The first thing we would like to clarify is that the sea ice phenology metrics plotted in the original Figure 4a were acquired from the Palmer LTER programme (i.e. the long-term monitoring programme run in the U.S. Antarctic Palmer Station), which monitors a large region within the southern-mid WAP (<https://pallter.marine.rutgers.edu/research-groups/research/>), stretching from 64°S to 70°S. Although we originally used this dataset since it is one of the most renowned and readily available sea ice datasets in the WAP, we have further inspected the data and it is now clear that the region does not accurately match the clusters we identified in this work. See the following Figure (Fig. R1) for a comparison between the Palmer LTER grid and the clusters delineated in our work:

[figure redacted]

Figure R1: Comparison between the Palmer LTER regional grid (left; extracted from the Palmer LTER website) and the clusters identified in our work (right; from Figure 2).

Therefore, we recalculated the sea ice phenology metrics, as well as sea ice extent, for each of the three clusters in the WAP where the changes in phytoplankton biomass and bloom phenology are occurring (DRA, BRS, and GES). The new Supplementary Figure 6 shows how the sea ice advance and retreat have changed since 1982 for the area of each cluster. We also added a regression line for the period since 2010-, to show how each metric has changed in recent years.

Sup. Fig. 6: Interannual variability of Sea Ice Advance (blue) and Retreat (red) dates for the regions of the WAP - DRA (a), BRS (b) and GES (c) - during 1980 to 2022. Full lines correspond to the linear regression for the 1982-2022 period, while the dashed lines correspond to the 2010-2022 period. These regions were chosen since they were the only ones that exhibited trends in biomass and/or bloom phenology (see Tables 1 and 2).

We can see that, for all clusters, the linear regression of the dates of sea ice advance (the start of the sea ice season; after the austral summer) since 2010 appears to follow the long-term trend (i.e. becoming later), even if only slightly. Nevertheless, we would like to stress that neither of the linear regressions are statistically significant, which is likely due to the high variability which characterizes sea ice phenology. This is in line with our results that show that phytoplankton biomass in the autumn appears to have increased in all three clusters (Table 1), as well as with the later bloom terminations observed in DRA and BRS (Table 2).

Interestingly, the linear regression of the dates of sea ice retreat (i.e. the end of the sea ice season; after the austral winter) for 2010 onwards show differences between the clusters. The northernmost regions (DRA and BRS) exhibit a trend towards earlier sea ice retreats, which follows the long-term trend that has long been described. In the southernmost, coastal region (GES), however, sea ice retreats appear to be becoming later. Although none of these trends are statistically significant, they appear to fit our results. For instance, we observed statistically significant increases in the biomass of spring months for both DRA and BRS, which are likely related to the earlier sea ice retreat and the positive effect it has on phytoplankton growth (Table 1).

Yet, we do not observe the same statistically significant increase in GES (Table 1). When writing the original version of the manuscript, we assumed this could be a consequence of the high temporal and spatial variability in chlorophyll-a that characterizes onshore waters of the WAP. However, it is possible that this may be due to a potential reversal of the long-term trends towards earlier sea ice retreat, as seen in Sup. Fig. 6c, although the lack of statistical significance prevents any definite conclusions. The GES region is also the closest to the Palmer LTER grid, overlapping with Palmer Station and the coastal systems sampled by the Palmer LTER programme. This overlap is coherent with the fact that, as pointed out by the reviewer, the sea ice retreat does not appear to become earlier since 2010 for the original Figure 4a, which uses Palmer LTER sea ice metrics. A recent study (Eayrs et al. 2021; Figure R2) also appears to support this distinction between northern and southern WAP, since it shows a recent (2015-2018) sea ice concentration decline in northern WAP (which matches the 1979-2015 trend), yet an increase in sea ice concentration towards the southern WAP. It is possible that this is related to the fact that atmospheric warming trends at the Antarctic Peninsula may have paused (or reversed, in some places) since 2000, although the overall consensus is that this is a case of natural interannual variability superimposed on the longer-term climate-change-driven trends (Turner et al., 2016).

[figure redacted]

Figure R2 (extracted from Eayrs et al. 2021): Variability in Sea Ice Concentration derived from passive microwave remote sensing. b, Mean annual SIC 1979–2018. c, Mean annual SIC changes from 1979 to 2015. d, Mean annual SIC changes from 2015 to 2018.

While we have included Supplementary Figure 6 in the Supplementary Material, we have also opted to redo and recontextualize Figure 4a (Line 525). Our original goal for Figure 4a was to offer context for the reader regarding the sea ice trends that have been described for the WAP. The new Figure 4a is now clearer, more complete, and more adequate for our work. It now presents sea ice extent data specifically for each of three subregions where impacts were observed (DRA, BRS, and GES). Along with the sea ice extent, the mean dates of sea ice retreat and advance are also shown. Furthermore, it unequivocally shows that, for all identified subregions, the 1998-2022 period is characterized by mean lower sea ice extent, earlier sea ice

retreat and later sea ice advance than the previous years between 1982-1997. We still redirect the reader in the main text to the new Supplementary Figure 6 when discussing the phenology of sea ice, but we think that the Figure 4a is more informative and clear this way. Sea ice phenology can be a limited metric, as it only indicates the first and last day of the sea-ice-free season in a given region. Therefore, it does not consider the extent of sea ice in that region nor the spatial distribution of sea ice, which can vary greatly along the Antarctic Peninsula coast due to the existence of bays, straits, fjords, as well as other sources of ice, such as glaciers and ice shelves.

New Figure 4a: Yearly average cycle of the Sea Ice Extent (103 km²) in each region (DRA, BRS, GES) between 1982-1997 (blue) 1998-2022 (red). For each period, vertical lines indicate the average date of sea ice advance (SIA) and sea ice retreat (SIR; dashed).

We have also included the following more recent references in the main text that report observations related to ice loss across the WAP: i) Lin et al. 2021 (decline in sea ice well into the 2000s); ii) Flexas et al. 2022 and Davison et al. 2024 (widespread increase in glacier runoff and melt rates); iii) Andreasen et al. 2023 (ice shelves decline by over 6000 km² between 2009 and 2019) (Lines 39, 193). The main text has been revised to accommodate for all the changes made to Figure 4a (e.g. Lines 193-195). We have also ensured that all comparisons of our results with ones from previous studies clearly indicate the studies' analysed time periods, thus offering better context to the reader (e.g. Lines 131-133, 156-157, 178).

Note that Figures R1 and R2 are exclusive to this point-by-point response and only serve to support our responses to the comments provided by the Reviewer 1.

In parenthesis, this reminds me very much of an Arctic Ocean paper that found similar phenological changes following the well-known sea ice changes that have happened in the Arctic Ocean, in case the authors did not know of it: Ardyna, M., M. Babin, M. Gosselin, E. Devred, L. Rainville, and J.-É. Tremblay (2014), Recent Arctic Ocean sea ice loss triggers novel fall phytoplankton blooms, *Geophys. Res. Lett.*, 41, 6207–6212, doi:10.1002/2014GL061047

At the moment, I suggest the authors to consider my suggestions and those of other reviewers, to improve the quality of their study for consideration of publication in Nature Communications.

I wish the authors the best of luck with the review process.

Best regards

Authors: We are grateful for the suggestion of this highly relevant paper and have cited it in the text as an analogous example of fall blooms in a polar region arising from climate change (Lines 235-238).

Specific comments:

Line 20: in this place, the authors mention that they are going to present their analyses over 25 years of data (1998-2022), but the final year of analyses is presented as 2021 in many other places of the manuscript. So I would suggest the authors to be very precise and also include the month for the start and end dates, since we are dealing here with Austral primary productivity seasons.

Authors: We agree that we must be clear about the periods analysed in our study. As mentioned earlier, we now provide all major information regarding the datasets utilized in the new Supplementary Table 3. Furthermore, we have carefully revised the text to prevent any confusion for the reader when comparing datasets with different temporal coverage (see our response to the first major comment).

Line 27: the end year for the “recent intensification of the Southern Annular Mode” is not given in this sentence.

Authors: Thank you for bringing this to our attention. We have added “ongoing” instead of specifying an end year, as the intensification of SAM is ongoing (Lines 28).

Lines 29-30: “region” is repeated twice in this sentence. Please remove one occurrence.

Authors: Removed.

Line 30: “raise important critical questions”. Please choose between “important” and “critical”, no need to use both adjectives in this sentence.

Authors: Done. Thank you.

Lines 34-38: In the first lines of the introduction, where the references #1, #2, #4 and #5 are used to describe the environmental changes that have happened in the Peninsula over the last

decades, the authors must bring more recent references of the physical changes that have occurred in the WAP since the most recent of these references (i.e. Montes-Hugo et al. 2009). I will come back to this question later, but I think it is essential to follow the argumentation of the authors that we also know what physical changes have happened in the WAP since these seminal studies were published.

Authors: We fully agree with the reviewer that more updated context should be provided regarding the entire period under study, for which we have added references to the following recently published works that convey the fact that the physical changes to sea ice are currently ongoing: Siegert et al. 2019, Lin et al. 2021, Flexas et al., 2022, Andreasen et al., 2023, Davison et al. 2024 (e.g. Lines 39, 193).

Line 85: when mentioning Figure 2 here, there also seems to be inconsistencies regarding the dates which are given as “1998-2022” in the Figures panels and as “1998-2021” in the Figure legend (line 450). Please correct throughout the manuscript.

Authors: Thank you, the analysed period for Figure 2 has been corrected to 1998-2022.

Lines 86-90: so does that mean that the other regions are less likely to experience iron limitation? The authors should comment on this in the paper and bring previous references.

Authors: Indeed, iron limitation is less likely to occur in the other regions due to their coastal nature, thus being more associated with higher sea ice coverage and surrounding glaciers. Glacial melt, in particular, serves as a key source of iron in the Antarctic Peninsula, thereby mitigating iron limitation during the austral summer. We have added one reference to ensure this is clear for the reader. (Annet et al. 2017; Lines 91).

Line 108: here the period stated is 1997-2021, contrary to what is indicated in the abstract. In the legend of Figure 3 (line 458), the period indicated is 1997-2022. And in the legend of Supplementary Figure 3 (line 16), the period indicated is also 1997-2022. The authors should be consistent throughout the paper regarding the exact period they ran their analyses on.

Authors: Thank you, we have uniformized the dates.

Line 119: “2011-2011” should be “2011-2021” I believe. However, the authors should make sure the period studied is consistent throughout the paper.

Authors: This has been corrected.

Lines 134-138: the authors’ Supplementary Figure 3b also suggests this increase in phytoplankton biomass in the Gerlache Strait in summer.

Authors: Added to the text (Lines 143-146).

Lines 150-151: this paragraph by the authors is about the increase in phytoplankton biomass in the northern WAP (BRS and DRA). However, this sentence and the Supplementary Figure 5 present data from the Southern (or mid-) WAP (i.e. Palmer Station). Therefore, this argumentation is not valid and the authors should present wind data trends for the whole WAP to feed their argumentation.

Authors: We used wind data from Palmer Station as an example of the increasing winds trend seen across the Western Antarctic Peninsula, including its northern sector. However, we agree that presenting wind data for different regions contribute to strengthen our discussion. Therefore, wind data from the Marambio (on the northern tip of the Antarctic Peninsula) and Great Wall (South Shetland Islands, in the Bransfield Strait) have now been added to the Supplementary Figure 5. All three show a similar statistically significant increasing trend in wind speed.

Lines 144-157: I think that the major problem in this paragraph and in other places of this paper is that the authors compare their long-term study (1998-2022) with other long-term studies that were performed using data from different periods. For example, the Montes-Hugo et al. (2009) study compares two very different decades (1978-1986 and 1998-2006) than the present paper. However, the physical dynamics of the WAP have not evolved linearly during the period presented in Montes-Hugo and the period presented in this paper. This is the case of sea ice phenology for example. Therefore, the present study is not comparable to these previous seminal studies and the authors should be more cautious when comparing their results with these previous studies.

Authors: We have already addressed this concern earlier (see the response to the second major comment). Nonetheless, we agree that directly comparing our results with the ones from Montes-Hugo et al. (2009), as well as any of the other studies covering different periods from ours, should be done carefully. Therefore, we have revised this part of the main text to be more prudent (e.g. Lines 125, 156-157).

Line 185: in the legend of Figure 4a, only the equation for the linear regressions for the sea ice advance is given. The authors should also give the equation for the linear regressions for the sea ice retreat which they also plot.

Authors: We have now redone Figure 4a, as explained in the second major comment.

In Figure 4, I think that this would be important to highlight the trends that correspond to the period that the authors use to study the change in phytoplankton biomass and phenology, so 1998-2022. This will be important when trying to interpret the role of sea ice phenology changes in phytoplankton dynamics. What happened in the 2 decades prior to 1998 is interesting but not relevant regarding the data presented in this paper.

Authors: We have now redone Figure 4a, as explained in the second major comment.

Line 187: I guess that the authors should also explain somewhere in the manuscript, perhaps here, that sea ice presence acts to decrease light penetration to the upper ocean. Even though relatively known, this may still be useful to some readers.

Authors: We have now added this information (Lines 218-220).

Line 191: The statement that sea ice declining rates are higher in the Gerlache Strait compared to the Drake Passage and Bransfield Strait needs to be backed up by showing relevant sea ice trends in the aforementioned regions or by presenting a reference that has shown this previously, if possible, a recent reference.

Authors: We have removed this sentence from the text.

Line 193: “while spring and summer biomass contribution slightly decreased...” add missing word.

Authors: Done. Thank you.

Line 194-196: since the authors only give the equation of the linear regression for sea ice advance, the reader cannot judge if this statement is true. Again, here, only the period 1998-2022 will be relevant to the present study.

Authors: The new Figure 4a, along with the Supplementary Figure 6, supports the statement made. We also would like to mention that several studies have shown that the rate at which sea ice advance is becoming later is higher than the rate at which sea ice retreat is becoming earlier (e.g. Stammerjohn et al. 2008; Moreau et al., 2015).

Line 201-204: while I agree with the authors’ statement that phytoplankton stocks are important here, I think that the authors should still consider and mention nutrients’ limitation for autumnal primary productivity.

Authors: We agree and have mentioned the potential influence of nutrient limitation on the autumn (Lines 225).

Line 204-206: as of now, this statement cannot be verified since the authors do not show the trend regarding sea ice retreat between 1998 and 2022, and instead use a much longer time series starting in 1980. This statement also contradicts the information the authors give in their Supplementary Figure 2.

Authors: We have removed this sentence to avoid confusing the reader.

Line 238: this is not entirely clear to me if the authors tested the correlation between the annual(?) SAM index and the annual(?) average Chl-a concentration for each pixel of the region, or they used an interannual(?) mean. I would think that the first is true but perhaps the authors could clarify.

Authors: We have mentioned in the Methods and in Line 263 that we tested the correlation between the mean September-April SAM index and the mean September-April Chl-a concentration for each pixel. Nevertheless, we have adjusted the sentence to make it clearer (Line 263).

Line 269: Should this sentence read “following strong El-Nino...”?

Authors: Correct. This has now been corrected.

Supplementary Figures:

Line 5: why is the period used for these data (1997-2021) different than the period used for the overall study (1998-2022)?

Line 10: why is the period used for these data (1997-2021) different than the period used for the overall study (1998-2022)?

Authors: This has been corrected now, see our response to the first major comment above.

Once again, we would like to thank Reviewer #1 for the insightful comments. We are confident that they have strengthened the quality and rigor of our paper. We also believe that the comments have contributed to the overall clarity of the manuscript.

Full list of the references included in this response:

- Andreasen, J.R., Hogg, A.E. and Selley, H.L., 2023. Change in Antarctic ice shelf area from 2009 to 2019. *The Cryosphere*, 17(5), pp.2059-2072.
- Davison, B.J., Hogg, A.E., Moffat, C., Meredith, M.P. and Wallis, B.J., 2024. Widespread increase in discharge from West Antarctic Peninsula glaciers since 2018. *EGU sphere*, 2024, pp.1-22.
- Flexas, M.M., Thompson, A.F., Schodlok, M.P., Zhang, H. and Speer, K., 2022. Antarctic Peninsula warming triggers enhanced basal melt rates throughout West Antarctica. *Science advances*, 8(31), p.eabj9134.
- Lin, Y., Moreno, C., Marchetti, A., Ducklow, H., O., Delage, E., Meredith, M., Li, Z., Eveillard, D., Chaffron, S. and Cassar, N., 2021. Decline in plankton diversity and carbon flux with reduced sea ice extent along the Western Antarctic Peninsula. *Nature communications*, 12(1), p.4948.
- Montes-Hugo, M., Doney, S.C., Ducklow, H.W., Fraser, W., Martinson, D., Stammerjohn, S.E. and Schofield, O., 2009. Recent changes in phytoplankton communities associated with rapid regional climate change along the western Antarctic Peninsula. *Science*, 323(5920), pp.1470-1473.
- Moreau, S., Mostajir, B., Bélanger, S., Schloss, I.R., Vancoppenolle, M., Demers, S. and Ferreyra, G.A., 2015. Climate change enhances primary production in the western Antarctic Peninsula. *Global change biology*, 21(6), pp.2191-2205.
- Siegert, M., Atkinson, A., Banwell, A., Brandon, M., Convey, P., Davies, B., Downie, R., Edwards, T., Hubbard, B., Marshall, G. and Rogelj, J., 2019. The Antarctic Peninsula under a 1.5 C global warming scenario. *Frontiers in Environmental Science*, 7, p.102.
- Stammerjohn, S.E., Martinson, D.G., Smith, R.C., Yuan, X. and Rind, D., 2008. Trends in Antarctic annual sea ice retreat and advance and their relation to El Niño–Southern Oscillation and Southern Annular Mode variability. *Journal of Geophysical Research: Oceans*, 113(C3).
- Turner, J., Lu, H., White, I., King, J.C., Phillips, T., Hosking, J.S., Bracegirdle, T.J., Marshall, G.J., Mulvaney, R. and Deb, P., 2016. Absence of 21st century warming on Antarctic Peninsula consistent with natural variability. *Nature*, 535(7612), pp.411-415.

Reviewer #2:

The paper “Phytoplankton trends in a changing climate: higher biomass and longer blooms in the West Antarctic Peninsula” by Ferreira and colleagues describes a satellite-based study of the changes in phytoplankton biomass and phenology in waters around the Antarctic Peninsula. Although the title refers to the west side of the peninsula, the east side was considered as well. The paper provides a nice description of the changes over the last few decades and provides reasonable explanations for them. I found the paper to be informative and interesting and their methods robust. I only have a few minor issues that the authors should consider when revising their manuscript. I describe them below.

Authors: We are grateful to Reviewer #2 for the thoughtful and constructive feedback. We will now address each of the minor issues raised.

Line 22. What is meant by climate change here? Is this referring to anthropogenic or natural climate changes? This should be made clear here.

Authors: We are referring to anthropogenic climate change. We have now added “anthropogenic” to this line to make it clear for the reader.

Line 26-27. Do the authors claim it is anthropogenic climate change or the SAM that is driving phytoplankton phenology? Or do they believe that anthropogenic climate change is driving the SAM? They need to clarify if these changes are part of a natural climate cycle or something induced by human activities.

Authors: We claim that the trends seen in phytoplankton phenology are a consequence of long-term environmental changes likely to be associated with anthropogenic action (e.g. ocean warming and decrease of sea ice along the Western Antarctic Peninsula). The multidecadal rising trend in SAM has also been linked to anthropogenic climate change (e.g. Fogt & Marshall, 2020). We have made this clearer in the abstract and in the main text (Lines 23, 256). Nevertheless, we sought to be prudent throughout the text when attributing our results to anthropogenic climate change since this is a very rapidly changing region, and it is always difficult to distinguish between natural and anthropogenic, particularly at shorter time scales.

Line 90-91. What differences are being referred to here? The authors have only described one subregion at this point.

Authors: This sentence has been rephrased (Lines 91-92).

Line 114. These numbers need more specific units. I assume that this is chlorophyll, but it needs to be specified.

Authors: It is indeed chlorophyll *a*. This has been corrected.

Line 127-128. Could these observations be related to later ice freeze-up?

Authors: Our results do suggest that long-term trends towards later sea ice advance (i.e. ice freeze-up) are related to higher production in the autumn (Lines 190-228).

Line 137-138. Is this increase statistically significant? If the cutoff for significance is 0.05, then it is not statistically significant, and the authors should not say that there was an increasing trend.

Authors: While the increase in December biomass in the GES region is not statistically significant at 0.05 (although its p-value is less than 0.1), we think that it is important to retain this observation in the manuscript. GES is a region that exhibits high seasonal and interannual variability in phytoplankton biomass, which can make achieving statistical significance harder. Despite the near-significant trend, there are several pieces of evidence suggesting an increase in summer phytoplankton biomass in GES: i) the average summer biomass between 2011-2020 appears to have increased compared to the previous decade (Fig. 2d); ii) pixel-wise increasing biomass trends were observed for the GES region, both during the summer period and from

September to April (Fig. 3a; Supplemental Fig. 3b). However, we agree with Reviewer #2 that we should be more accurate, for which it is now mentioned in the sentence that the observation is not statistically significant using a p-value threshold of 0.05 (Lines 143; p-value = 0.094 and R=0.34).

Line 152-153. Has a long-term deepening of the mixed layer been observed, or is this only assumed from the stronger winds?

Authors: In this study, we do not present mixed layer data because we would need the physical profile (in situ temperature and practical salinity) to estimate the seawater potential density and, subsequently, the mixed layer depth. However, most of the open access data available for this region do not present physical profiles, making it unfeasible to calculate and consequently use this parameter. Nevertheless, we can assume this based on the wind trends (Supplementary Figure 5), as the deepening of mixed layer in open ocean regions is strongly driven by wind patterns. For the WAP, several studies have already reported the deepening of the mixed layer depth associated with strong winds (e.g., Schofield et al. 2018).

Line 188-189. I'm not sure what generalized means. Is it that phytoplankton growth was the same throughout the WAP?

Authors: We have changed 'generalized' to 'widespread'.

Line 209-211. Do nutrients not play a role at all? Eventually, the iron will be consumed and the bloom will end, unless there are mechanisms to mix new iron to the surface. A longer growing season will not necessarily guarantee higher production. Plus, it is new production that is important to the ecosystem. Continuing a bloom for longer on recycled nutrients provides fewer benefits for the ecosystem and does not sequester carbon.

Authors: We agree that nutrients can play a significant role in limiting phytoplankton growth. We see this in the DRA region, which is the most open ocean region in our study. While we observe a longer growing season in the DRA, the summer bloom ends relatively early (mid-February, on average), most likely due to nutrient limitation. However, based on previous studies, macronutrient availability in more coastal regions, such as the BRS, can still be high in late February and March, potentially fuelling phytoplankton growth in the early autumn (e.g. Monteiro et al. 2023). In addition, onshore waters of the WAP have also been described as highly abundant in iron, even in late summer and autumn (Annett et al. 2017; Sherrell et al. 2018; Pan et al. 2020).

Line 260. Maybe change "compartments" to "components" ?

Authors: We are grateful for the suggestion. However, in this case, we think that the use of compartments (i.e. any of the enclosed parts into which a space (...) is divided) is more appropriate than components in this context (i.e. one of the parts of a system, process, or machine). Definitions were taken from the Cambridge Dictionary website.

Line 268-270. If it is important to consider ENSO, then why hasn't it been considered in this study?

Authors: While ENSO can indeed play a significant role in shaping the interannual variation in the coastal WAP (e.g. summers following strong El-Niño may yield large blooms; Costa et al. 2020, 2021), our study focuses on the long-term trends in phytoplankton, with SAM being typically regarded as the key climate mode linked with long-term environmental change in this region. In addition, it is important to highlight that the effects of the ENSO are still not well understood for the WAP, especially the delay of ENSO effects reaching the WAP after its initial expression in the Equatorial Pacific Ocean. Nonetheless, we have revised the sentence to make clear for the reader that ENSO is likely to hold a more interannual importance (Lines 296-299).

Line 272-273. Wouldn't the bigger issue be the decline in krill and consequently the animals that rely on them for food?

Authors: Of course, we agree that a decline in krill can be devastating for the Antarctic food web. In this sentence, we meant to stress how important the relationship between phytoplankton and krill is and how a change in size composition of phytoplankton may affect phytoplankton predation by krill. We have revised the text to make this clearer (Lines 300-305).

Line 510. Insert space (5 km)

Authors: Done.

Line 512. Change "photosynthetic" to ""photosynthetically

Authors: Done.

Line 513. Should be "data were"

Authors: Done.

Line 518. What is the sea ice concentration cutoff used to separate the sea ice season from the open water season? It may be described later, but should be given here first.

Authors: The sea ice concentration cutoff is 15%. The day of sea ice advance (first day of the sea-ice season) is the first day where sea ice concentration is higher than 15% for at least 5 consecutive days. The day of sea ice retreat is the inverse, i.e., the first day after sea ice season when the sea ice concentration drops below 15%. All details can be found Lines 579-588, in the Methods.

Line 542. Insert space (10 km)

Authors: Done.

Line 551. Insert space (4 km)

Authors: Done.

Reference list:

- Annett, A.L., Fitzsimmons, J.N., Séguret, M.J., Lagerström, M., Meredith, M.P., Schofield, O. and Sherrell, R.M., 2017. Controls on dissolved and particulate iron distributions in surface waters of the Western Antarctic Peninsula shelf. *Marine Chemistry*, 196, pp.81-97.

- Costa, R.R., Mendes, C.R.B., Tavano, V.M., Dotto, T.S., Kerr, R., Monteiro, T., Odebrecht, C. and Secchi, E.R., 2020. Dynamics of an intense diatom bloom in the Northern Antarctic Peninsula, February 2016. *Limnology and Oceanography*, 65(9), pp.2056-2075.
- Costa, R.R., Mendes, C.R.B., Ferreira, A., Tavano, V.M., Dotto, T.S. and Secchi, E.R., 2021. Large diatom bloom off the Antarctic Peninsula during cool conditions associated with the 2015/2016 El Niño. *Communications Earth & Environment*, 2(1), p.252.
- Fogt, R.L. and Marshall, G.J., 2020. The Southern Annular Mode: variability, trends, and climate impacts across the Southern Hemisphere. *Wiley Interdisciplinary Reviews: Climate Change*, 11(4), p.e652.
- Monteiro, T., Henley, S.F., Pollery, R.C.G., Mendes, C.R.B., Mata, M., Tavano, V.M., Garcia, C.A.E. and Kerr, R., 2023. Spatiotemporal variability of dissolved inorganic macronutrients along the northern Antarctic Peninsula (1996–2019). *Limnology and Oceanography*, 68(10), pp.2305-2326.
- Pan, B.J., Vernet, M., Manck, L., Forsch, K., Ekern, L., Mascioni, M., Barbeau, K.A., Almandoz, G.O. and Orona, A.J., 2020. Environmental drivers of phytoplankton taxonomic composition in an Antarctic fjord. *Progress in oceanography*, 183, p.102295.
- Schofield, O., Brown, M., Kohut, J., Nardelli, S., Saba, G., Waite, N. and Ducklow, H., 2018. Changes in the upper ocean mixed layer and phytoplankton productivity along the West Antarctic Peninsula. *Philosophical Transactions of the Royal Society A: Mathematical, Physical and Engineering Sciences*, 376(2122), p.20170173.
- Sherrell, R.M., Annett, A.L., Fitzsimmons, J.N., Rocanova, V.J. and Meredith, M.P., 2018. A 'shallow bathtub ring' of local sedimentary iron input maintains the Palmer Deep biological hotspot on the West Antarctic Peninsula shelf. *Philosophical Transactions of the Royal Society A: Mathematical, Physical and Engineering Sciences*, 376(2122), p.20170171.
- Stammerjohn, S.E., Martinson, D.G., Smith, R.C., Yuan, X. and Rind, D., 2008. Trends in Antarctic annual sea ice retreat and advance and their relation to El Niño–Southern Oscillation and Southern Annular Mode variability. *Journal of Geophysical Research: Oceans*, 113(C3).

REVIEWERS' COMMENTS

Reviewer #1 (Remarks to the Author):

Review of “Phytoplankton trends in a changing climate: higher biomass and longer blooms in the West Antarctic Peninsula” submitted to Nature Communications by Ferreira et al.

I read the new version of the paper and the response to the reviewers’ comments by the co-authors, and I am satisfied with the authors' response. They have addressed all the comments satisfyingly and improved the robustness of the paper, especially regarding the timing of the trends they observe and compared to previously published results.

I find the results of this paper interesting and contributing significantly to our knowledge of this climate-sensitive and important region of the Southern Ocean. The paper is well written and interesting. Therefore, I recommend the paper for publication in Nature Communications.

I congratulate the authors on a nice and interesting paper.

Best regards

Specific comments:

Line 262: “Tagliabue et al.” the last name is miss-spelled here

Reviewer #2 (Remarks to the Author):

The authors have adequately addressed all of my concerns.